# An Improved Spanning Tree-Based Algorithm for Coverage of Large Areas Using Multi-UAV Systems

Jan Chleboun [1], Thulio Amorim [2], Ana Maria Nascimento [3] and Tiago P. Nascimento [1,3,*]

[1] Department of Cybernetics, Czech Technical University in Prague (CTU), 121 35 Prague, Czech Republic
[2] Natalnet Associate Laboratories, Universidade Federal do Rio Grande do Norte (UFRN), Natal 59078-970, RN, Brazil
[3] Laboratory of Systems Engineering and Robotics (LASER), Department of Computer Systems, Universidade Federal da Paraíba (UFPB), João Pessoa 58055-000, PB, Brazil
[*] Correspondence: tiagopn@ci.ufpb.br or pereiti1@fel.cvut.cz; Tel.: +55-83-99911-5577

**Abstract:** In this work, we propose an improved artificially weighted spanning tree coverage (IAW-STC) algorithm for distributed coverage path planning of multiple flying robots. The proposed approach is suitable for environment exploration in cluttered regions, where unexpected obstacles can appear. In addition, we present an online re-planner smoothing algorithm with unexpected detected obstacles. To validate our approach, we performed simulations and real robot experiments. The results showed that our proposed approach produces sub-regions with less redundancy than its previous version.

**Keywords:** robot exploration; spanning tree; UAV; trajectory planning

## 1. Introduction

Drones (unmanned aerial vehicles (UAVs)) have gained popularity in recent years due to their ability to perform a diverse range of tasks. Environmental surveillance [1], object transportation [2], and historical site documentation [3], are among such tasks. A common commercial UAV is a small, lightweight helicopter with a minimum of four propellers (see Figure 1). The advantage of a UAV is the capability of an agile flight and a small size. In contrast, the biggest weakness is its short autonomy and its limited computational capability. Frequently, applications tend to use multiple UAVs.

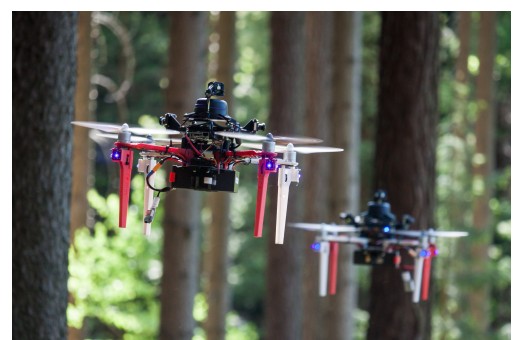

**Figure 1.** An example of multi-rotor UAV platform [4].

In multi-robotics, a commonly encountered problem in various applications consists of determining an optimal path involving all points of a given area of interest while avoiding sub-regions with specific characteristics, such as obstacles or no-fly zones [5]. One of these problems is the coverage path planning (CPP) problem. It consists of covering a large area through the application of path-planning approaches that minimize time-of-flight. One technique often employed is the use of a multi-robot system.

To deal with more sophisticated requirements, multi-robot systems with proper path planning provide a promising solution that can significantly enhance effectiveness and robustness when compared with a single-robot system [6]. In contrast, addressing the CPP problem is more difficult when working with multiple robots [5].

The work of Huang et al. [7] explained that although the state-of-the-art literature presents different multi-robot coverage path planning (MCPP) algorithms to solve the coverage path planning (CPP) problem of robots in specific areas, in outdoor environments, complex geographic environments reduce the task execution efficiency of robots. The same authors further explain that existing CPP approaches have hardly considered environmental complexity, while other studies have considered robot applications but had a small study area and a single cover type, such as a sweeping robot. Thus, robots are always assumed to have a constant visual field and moving speed in the entire area in existing multi-robot coverage path planning (MCPP) algorithms. The performance efficiency of robots in emergency search and rescue tasks is seriously affected due to the complexity and variation of the environmental surface, so a robot may have different visual fields and moving speeds in different environments. In the case of this study, we tackled not only a large area with a considered variation of the environment complexity but also the speed of the UAVs and the implications of that on the resulting trajectory.

A common approach to deal with the MCPP problem is the spanning tree coverage (STC) algorithm. This approach guarantees an optimal covering path in linear time when certain conditions are met [5]. Despite the impressive contributions from previous works [5,6,8], the STC algorithms are still an active research topic with promising development opportunities. In this work we proposed an improved artificially weighted spanning tree coverage (IAWSTC) approach designed to minimize the redundancy regions for the coverage area of the multi-UAV system and to enable the UAVs to handle unexpected obstacles. Thus, the main contributions are:

1. an Overlapping Region Reduction approach designed to minimize the redundancy regions for the coverage area;
2. a trajectory smoothing approach based on last square solutions that we call trajectory smoothing using least squares solution (TSULSS). This approach enables a fast smoothing of long trajectories;
3. An obstacle avoidance and replanning approach for unexpected obstacles (i.e., moving obstacles or fixed obstacles that exist but are unknown due to the absence of a map).

## 2. Related Works

In 2006, Agmon et al. [9] presented one of the early works that tried to solve the CPP problem. The authors had proposed to construct spanning trees. The drawback of their approach was the coverage of the desired regions was not guaranteed to be performed more than once, causing redundancy on the planned path.

Many other approaches for the CPP problem have been proposed. Some approaches use a single robot, others multiple robots. Methods that use a single robot to solve the CPP problem used geometric approaches such as the decomposed polygon [10] and polygonal environments with geometric vector algorithm [11], game-theory approaches [12], Dubins traveling salesman problem [1], and so on. Most of the approaches in the literature either perform only simulations or do not consider obstacles or uncertainties.

When investigating multi-robot system approaches to solve the CPP problem, an example is the work of Avellar et al. [13]. When involving a multi-robot system to solve the CPP, the problem of CPP is then expanded to a MCPP problem. In their work, Avellar et al. [13] adopted a more practical perspective, taking into account environmental factors that exist in real-world scenarios such as maximum time-of-flight, setup time, and the state of the battery of the robot. In addition, Kapoutsis et al. [5] presented an approach to deal with the path planning problem of a team of mobile robots, to cover an area of interest with prior-defined obstacles. The proposed technique transformed the original

MCPP into several single-robot CPP problems, the solutions of which constitute the optimal MCPP solution, alleviating the original MCPP explosive combinatorial complexity.

Recently, Dong et al. [6] extended the work of Kapoutsis et al. [5], proposing the artificially weighted spanning tree coverage (AWSTC) algorithm as a suitable solution for multi-UAV area coverage in a distributed manner. The algorithm decomposes the environment in cells. The robots take turns and insert a new cell into their coverage subregion iteratively, using a formula to assign a score to each cell. This formula tries to ensure that each robot visits the uncovered cells while avoiding cells already explored by its partners. After this, a STC algorithm converts the subregion into a trajectory. Since the generated path contains many rough turns, a trajectory smoothing algorithm minimizes this issue using Bézier curves. In addition, the authors performed experiments with real robots in a controlled environment, with no unexpected obstacles.

In the AWSTC, the environment is divided, such that:

$$C_u = \{c_i | \, c_i \in C, c_i \cap A = \varnothing, c_i \cup O = \varnothing\}, \tag{1}$$

in which $A$ is an obstacle occupied area $O$ divided into equal-sized cells $C$.

When using multiple robots, the coverage problem involves finding a suitable path for each agent such that:

$$\min \max \|P_j\|,$$
$$\text{s.t.} \bigcup_{j=1}^{N} P_j = C_u, \tag{2}$$

in which the $P_j$ is the obtained path for the $j$th robot, and $N$ is the number of robots.

As indicated in Equation (2), to ensure efficient coverage, the algorithm must minimize the lengths of the resulting trajectories. However, the AWSTC approach ends up with several redundancies within the calculated trajectory and does not take re-planing into account. Furthermore, the smoothing algorithm the authors used could be further improved.

Thus, we modified the algorithm of Dong et al. [6] aiming to minimize the redundancy of the calculated trajectories. We also focused on a decentralized version that ensures equal distribution of computational load and trajectory length. Finally, we treated the unexpected obstacle problem while smoothing the trajectories to be performed by the UAV.

## 3. Problem Formulation

Generally, the abstraction of the problem depends on the sensors and actuators installed on the robot. However, we assumed that the area to be covered is inside a 2D rectangle, which we can discretize into a finite set of cells with equal size. The position of each obstacle inside the area is known, and the robot can accurately locate itself, traveling from its current location to any unblocked adjacent cell without any motion uncertainty [5]. The environment division and its terminologies can be seen in Figures 2 and 3.

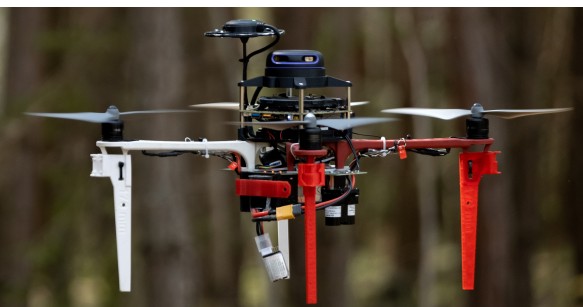

**Figure 2.** An agent is a UAV performing the desired task.

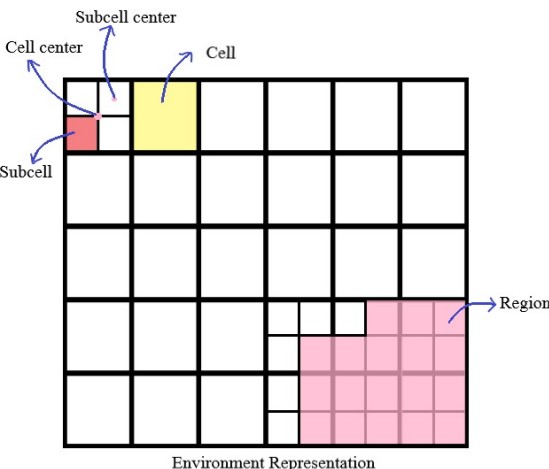

**Figure 3.** Environment and grid definitions.

For ease of understanding, we formally added the required terminology and definitions for accurate descriptions of the algorithms.

**Definition 1.** *An environment is a workspace used to perform the coverage task and the grid division of the environment is called environment representation. The objective of our approach is to perform a full coverage of the environment.*

**Definition 2.** *A cell is considered as explored after a robot passes above its center or above each sub-cell center that is part of the respective cell. A cell is unexplored if it is not visited by the robot.*

**Definition 3.** *A trajectory is defined as a path that is composed of a sequence of points in a 2D workspace.*

**Definition 4.** *The Manhattan distance of cells $C_1, C_2$ is a metric defined as:*

$$D_{C_1,C_2} = |x_{C_1} - x_{C_2}| + |y_{C_1} - y_{C_2}|, \tag{3}$$

*where $x_{C_1}, y_{C_1}$ are the first cell center coordinates, $x_{C_2}, y_{C_2}$ are the second cell center coordinates, and $D_{C_1,C_2}$ is the resulting Manhattan distance.*

**Definition 5.** *Two cells $C_1, C_2$ are neighbor cells if:*

$$D_{C_1,C_2} = CS, \tag{4}$$

*where CS is the side length of a cell.*

**Definition 6.** *Two $C_1, C_2$ cells are diagonal neighbor cells if:*

$$d_{C_1,C_2} \leq CD, \tag{5}$$

*where CD is the diagonal length of a cell. For every adjacent cell, there is a diagonal neighbor cell, but not the reverse.*

**Definition 7.** *The status of exploration of a cell is the binary information on whether the cell is explored (i.e., either already visited by the robot or containing an obstacle), or unexplored.*

## 4. Improved Artificially Weighted Spanning Tree Coverage

We addressed the MCPP problem in the same way as Dong et al. [6], using a STC algorithm. As formally described in Section 3, the main objective of the coverage process is to divide the environment into a workspace represented by subregions. These subregions must be covered by each robot. Finally, each robot generates its trajectories such that each subregion is fully explored.

### 4.1. Partition of the Environment

In this step, each robot selects its subregion of exploration $S$ in a distributed manner. Iteratively, the robots take turns evaluating all unexplored cells, choosing the cells with the highest calculated scores. Then, these cells are included in the subregion $S$ and marked as explored. This process repeats until all cells in the workspace are explored. We outline the complete process in Algorithm 1. To improve the behavior of the previous environment partitioning [6], we modified the formula used to calculate the score of the suitable cells. The original evaluation formula taken over from the previous work [6] consists of three terms $E_j^a$, $E_{j,k}^b$, and $E_j^c$. Here, the $i$th robot calculates the score $E_{i,j}$ of the $j$th cell through the equation below.

---

**Algorithm 1** Partition of the Environment

---

Create a workspace by performing a cell $C_j$ division of the environment
**for** each $C_j$, $j \in [1, 2, \ldots, M]$ **do**
    $C_j \leftarrow unexplored$
**end for**
Initialize subregions $S_i$
**for** each subregion $S_i$, $i \in [1, N]$ **do**
    $S_i \leftarrow i$th starting point of the UAVSP$_i$
    SP$_i \leftarrow explored$
**end for**
**while** *unexplored* and $C_i$ exists **do**
    **for** each UAV$_i$, $i \in [1, N]$ **do**
        Analyze if each cell with one or more adjacent cells belongs to the region $S_i$, by using Equation (10)
        find $C_{best} = \arg_j \max E_{i,j}$
        Add $C_{best}$ to $S_i$
        $C_{best} \leftarrow explored$
    **end for**
**end while**

---

$$E_{i,j} = E_j^a + \sum_{k=1,k\neq i}^{N} E_{j,k}^b + E_j^c. \tag{6}$$

where $E_j^a$ is the distance between the visited cell and the center of the unexplored workspace.

$E_j^a$ is used to ensure that cells that are located on the border of the unexplored workspace are given higher scores. $E_j^a$ can be calculated as:

$$E_j^a = \sigma_a \left( |x_i - x_{ce}| + |y_i - y_{ce}| \right), \tag{7}$$

in which $x_i, y_i$ are coordinates of the center of the cell currently being explored, $x_{ce}, y_{ce}$ are the coordinates of the center of the environment (a fixed referential), and $\sigma_a > 0$ is a positive constant used as a function gain.

Furthermore, $E_{j,k}^b$ is the distance between the *j*th cell and the region $S_k$ belonging to the *k*th agent. It increases the score of cells that are distant from the other UAVs, and is given as:

$$E_{j,k}^b = \sigma_b \min_{[x_j,y_j] \in S_k} \left(|x_j - x_i| + |y_j - y_i|\right), \tag{8}$$

in which $x_j, y_j$ are the region $S_k$ cell coordinates that belongs to the *k*th agent, and $\sigma_b > 0$ is a positive constant used as a function gain.

Finally, the last term $E_j^c$ specifies the status of exploration of the *j*th cell. It decreases the score of visited cells, and it can be calculated as:

$$E_j^c = \begin{cases} 0 & \text{if the cells are } \textit{unexplored} \\ -\mathcal{E}_M & \text{if the cells are } \textit{explored}, \end{cases} \tag{9}$$

in which $\mathcal{E}_M > 0$ is a large positive constant value.

*4.2. Proposed Approach*

Thus, here we propose modifications on the AWSTC algorithm, such that the score calculated for the *i*th cell to be visited by the *j*th agent will now, be estimated as:

$$E_{i,j} = E_j^a + \sum_{k=1,k\neq i}^{N} E_{j,k}^b + E_{j,m}^c + E_{i,j}^d, \tag{10}$$

where $E_{j,m}^c$ is a penalty term used to avoid redundancy in the coverage, and $E_{i,j}^d$ is a positive term applied to reward cluttered regions.

Note that the first and second terms, $E_j^a$ and $E_{j,k}^b$ respectively, remain the same from the previous work [6]. However, the third term $E_{j,m}^c$, which is responsible for decreasing the score of already explored cells, was modified. More formally:

$$E_{j,m}^c = \begin{cases} 0 & \text{for unexplored cells} \\ -\mathcal{E}_M \, D_{j,u} & \text{for explored cells}, \end{cases} \tag{11}$$

where $D_{j,u}$ is the Manhattan distance between the $C_j$ cell and the closest unexplored cell.

Our approach starts by analyzing the unexplored cells first, where $E_{j,m}^c = 0$. Furthermore, we included a new reward term $E_{i,j}^d$ that increases the score of cells from clusters. In other words, we estimate how many diagonal adjacent cells of the *j*th cell are found in the same region. More precisely:

$$E_{i,j}^d = \sigma_d \, \text{NC}_{i,j}, \tag{12}$$

where the number of diagonal adjacent cells of the cell $C_j$ that belong to the region $S_i$ corresponding to the *i*th robot is $\text{NC}_{i,j}$, and $\sigma_d > 0$ is a positive constant used as a function gain.

Our proposed approach also adds a mechanism to minimize the overlapping of different regions, while counting the cells similarly. Every cell $C_j$ that belongs to two or more regions is removed from every $S_i$ region if it does not violate the continuity of $S_i$. Then, if the cell $C_j$ does not belong to any region, it is listed as unexplored. Furthermore, the robots take turns analyzing all unexplored cells using Equation (10) and picking the highest-scored cell. At every turn, the respective robot will be the one in which the corresponding region has the smallest set of cells. The algorithm repeats this process until no solvable conflicts emerge. We summarize the entire process in Algorithm 2.

After the environment partitioning, the agent creates a trajectory that includes all cells of its subregion by using STC. By combining the partition of the environment algorithm with STC, we obtain the same AWSTC previous algorithm described [6]. However, if

we instead combine STC with our modified version of the partition of the environment algorithm, we obtain our proposed IAWSTC algorithm.

---

**Algorithm 2** Reduction of Overlapping Regions

---

**for** each cell $C_j$, $j \in [1, M]$ **do**
    **if** $\mathrm{NR}_{C_j} > 1$, where $\mathrm{NR}_{C_j}$ is the number of *regions* $S_i$, $i \in [1, N]$ which satisfy $C_j \in S_i$
**then**
        **for** each *region* $S_i$, $C_j \in S_i$ **do**
            **if** $S_i \setminus C_j$ is *continuous* **then**
                remove $C_j$ from $S_i$
            **end if**
        **end for**
        **if** $\mathrm{NR}_{C_j} = 0$ **then**
            $C_j \leftarrow$ *unexplored*
        **end if**
    **end if**
**end for**
**while** *unexplored* $C_j$ exists **do**
    $i \leftarrow$ *agent* with the least cells in its region $S_i$
    Analyse each cell with one or more adjacent cells that belong to region $S_i$ using Equation (10) as viewed by agent *i*
    find $C_{best} = \arg_j \max E_{i,j}$
    Add $C_{best}$ to $S_i$
    $C_{best} \leftarrow$ *explored*
**end while**

---

*4.3. Complexity*

Both the classic and improved versions of the partition of the environment (Algorithm 1) algorithm's time complexity is $\mathcal{O}(N^2)$. In addition, the STC time complexity is $\mathcal{O}(N)$. Therefore, as a result, we have a $\mathcal{O}(N^2)$ for both the AWSTC by Dong et al. [6] and our proposed IAWSTC.

**5. Trajectory Smoothing Using Least Squares Solution**

The trajectories generated by both algorithms contain many rough turns. Generally, a smoothing algorithm attenuates these turns to enable the controller of the UAV to track the resulting path. Note here that we do not tackle the well-known "smoothing problem" [14,15], which is the problem of estimating an unknown probability density function recursively over time using incremental incoming measurements. In the path planning field of research, there is another definition of "smoothing problem", which is the problem of generating polyline segments after the initial path points are obtained (or generated) from a path planning algorithm (such as A*, RTT, and so on) [16–18]. Thus, one must consider that the UAV usually deviates from the given reference due to the controller's inability to perfectly follow the desired trajectory, especially when there is a turn exceptionally sharp. Thus, a smoothing algorithm is decisive in keeping the UAV closer to the planned path and exploring the environment adequately.

Several trajectory smoothing approaches have been proposed in the past and they can be divided into three classifications: special curves, interpolation-based, and optimization [19]. The interpolation-based methods use polynomial interpolation [20], Bézier curves [21], cubic splines [22], B-splines [23], or NURBS curves [24]. Special curves usually use Dubin's curves [25], Clothoid [26], Hypocycloid [27], and other types of curves [19]. Optimization methods treat trajectory smoothing as a problem in which energy, time of execution, and other criteria are minimized.

In this work, we propose, along with other contributions, a smoothing algorithm that aims for simplicity and low computational cost, while going through each of the original

trajectory points. The final smoothened trajectory is sent to the UAV control system that is composed of an MPC tracker and a SE(3) controller. This control pipeline was proposed by our group in the past [4]. It is interesting to note that the MPC tracker checks the feasibility of the given trajectory based on the UAV dynamics. However, our smoothing approach "easy up" the computational processing for this tracker since the smoothing approach already gives a more feasible path for the UAV to follow. Thus, the visit to all points must be ensured in order to explore all the desired cells. Thus, we proposed the TSULSS. Our approach is based on the addition of equally distant $p$ points between every couple of points from the original trajectory (see Figure 4). Then, we minimize the function of the curvature of the entire new trajectory $\tau$. For this function, we use the sum of squared angles between pairs of consecutive vectors $v_i, v_{i+1} \in \tau$. Nevertheless, we are not able to formulate this function as the least squares solution of a system of linear equations. Thus, we decompose the vectors $v_i, v_{i+1}$ into three consecutive points $P_i, P_{i+1}, P_{i+2}$, while the distance $d_{mid,i}$ from the middle point $P_{i+1}$ to the center of mass of the three points are used to describe the angle between the two vectors $v_i, v_{i+1}$. Finally, we can describe the function of the curvature of $\tau$ as the sum of the squared distances $d_{mid,i}$ for every three consecutive points from the resulting path $\tau$. This function is calculated as:

$$f(\tau) = \sum_{i=0}^{N-2} \left[ \left( \frac{P_{i,x} + P_{i+1,x} + P_{i+2,x}}{3} - P_{i+1,x} \right)^2 \right.$$

$$\left. + \left( \frac{P_{i,y} + P_{i+1,y} + P_{i+2,y}}{3} - P_{i+1,y} \right)^2 \right]. \quad (13)$$

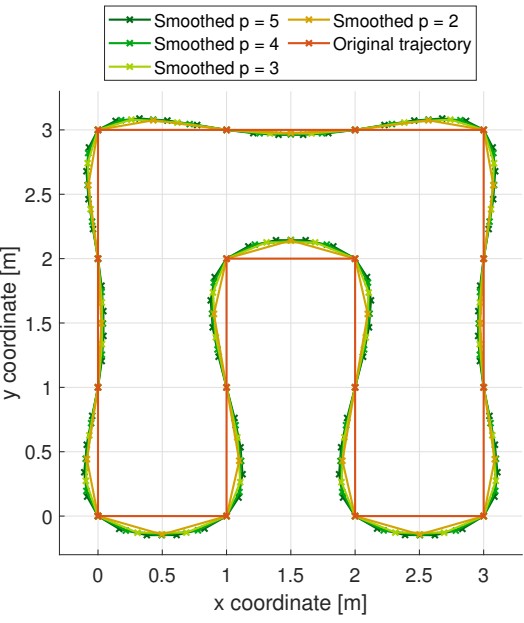

**Figure 4.** The meaning of the $p$ parameter—with higher values of $p$, more points are inserted into the trajectory, which leads to smoother turns.

The equation of the trajectory curvature $\tau$ is presented by Equation (13) below. However, one must also consider the shift of individual points. Thus, for each $P_i$ point, we replace it with $P_i + \lambda_i \delta_i$, where $\lambda_i$ is a binary variable that ensures the points of the original trajectory will not be changed during the smoothing process, and therefore, the UAV will explore all the cells, $\delta_i$ is the change of the $i$th point of the trajectory, and

$$\lambda_i = \begin{cases} 0 & P_i \text{ comes from the original trajectory} \\ 1 & P_i \text{ added on the first phase.} \end{cases}$$

The final function can be seen in Equation (14).

$$f(\tau) = \sum_{i=0}^{N-2} \left[ \left( \frac{P_{i,x} + \lambda_i\, \delta_{i,x}}{3} - 2\, \frac{P_{i+1,x} + \lambda_{i+1}\, \delta_{i+1,x}}{3} + \frac{P_{i+2,x} + \lambda_{i+2}\, \delta_{i+2,x}}{3} \right)^2 + \right.$$
$$\left. \left( \frac{P_{i,y} + \lambda_i\, \delta_{i,y}}{3} + 2\, \frac{P_{i+1,y} + \lambda_{i+1}\, \delta_{i+1,y}}{3} + \frac{P_{i+2,y} + \lambda_{i+2}\, \delta_{i+2,y}}{3} \right)^2 \right]. \quad (14)$$

$$f_f(\tau) = f(\tau) + \left( \frac{P_{N-1,x} + \lambda_{N-1}\, \delta_{N-1,x}}{3} - 2\, \frac{P_{1,x} + \lambda_1\, \delta_{1,x}}{3} + \frac{P_{2,x} \lambda_2\, \delta_{2,x}}{3} \right)^2 +$$
$$\left( \frac{P_{N-1,y} + \lambda_{N-1}\, \delta_{N-1,y}}{3} - 2\, \frac{P_{1,y} + \lambda_1\, \delta_{1,y}}{3} + \frac{P_{2,y} \lambda_2\, \delta_{2,y}}{3} \right)^2. \quad (15)$$

$$f_f(\tau) = \| \mathbf{A}\, \mathbf{x} - \mathbf{b} \|^2, \quad (16)$$

where

$$\mathbf{A} = \begin{bmatrix} \frac{\lambda_1}{3} & 0 & -\frac{2\lambda_2}{3} & 0 & \frac{\lambda_3}{3} & 0 & 0 & \dots & \dots & \dots & \dots & \dots & 0 \\ 0 & \frac{\lambda_1}{3} & 0 & -\frac{2\lambda_2}{3} & 0 & \frac{\lambda_3}{3} & 0 & \dots & \dots & \dots & \dots & \dots & 0 \\ \vdots & \vdots & \vdots & \vdots & \vdots & \vdots & \ddots & \vdots & \vdots & \vdots & \vdots & \vdots & \vdots \\ 0 & \dots & \dots & \dots & \dots & \dots & 0 & \frac{\lambda_{N-2}}{3} & 0 & -\frac{2\lambda_{N-1}}{3} & 0 & \frac{\lambda_N}{3} & 0 \\ 0 & \dots & \dots & \dots & \dots & \dots & \dots & 0 & \frac{\lambda_{N-2}}{3} & 0 & -\frac{2\lambda_{N-1}}{3} & 0 & \frac{\lambda_N}{3} \\ -\frac{2\lambda_1}{3} & 0 & \frac{\lambda_2}{3} & 0 & \dots & \dots & \dots & \dots & 0 & \frac{\lambda_{N-1}}{3} & 0 & \dots & 0 \\ 0 & -\frac{2\lambda_1}{3} & 0 & \frac{\lambda_2}{3} & 0 & \dots & \dots & \dots & \dots & 0 & \frac{\lambda_{N-1}}{3} & 0 & 0 \end{bmatrix},$$

$$\mathbf{x} = \begin{bmatrix} \delta_{1,x} & \delta_{1,y} & \delta_{2,x} & \delta_{2,y} & \delta_{3,x} & \delta_{3,y} & \dots & \delta_{N,x} & \delta_{N,y} \end{bmatrix}^T,$$

$$\mathbf{b} = \begin{bmatrix} -\frac{1}{3} P_{1,x} + \frac{2}{3} P_{2,x} - \frac{1}{3} P_{3,x} \\ -\frac{1}{3} P_{1,y} + \frac{2}{3} P_{2,y} - \frac{1}{3} P_{3,y} \\ \vdots \\ -\frac{1}{3} P_{N-2,x} + \frac{2}{3} P_{N-1,x} - \frac{1}{3} P_{N,x} \\ -\frac{1}{3} P_{N-2,y} + \frac{2}{3} P_{N-1,y} - \frac{1}{3} P_{N,y} \end{bmatrix}.$$

For trajectories in which $P_1 = P_N$, we added another term to round off the connection between the beginning and the end of the trajectory. The resulting function to be minimized can be seen in Equation (15). Note that we can rewrite this equation in matrix form for better clarity, as presented in Equation (16). Thus, our optimization problem is formulated as:

$$\min_{x \in \mathcal{R}^{2N}} \|\mathbf{A}\,\mathbf{x} - \mathbf{b}\|^2. \tag{17}$$

Finally, to improve the UAV control capability over the trajectory resulting from our approach, we used the Tikhonov regularization to penalize significant changes and introduced a $\mu$ parameter that set up the level of the smoothing that we desired on the trajectory. Thus, the final function is given by:

$$\min_{x \in \mathcal{R}^{2N}} \left( \|\mathbf{A}\,\mathbf{x} - \mathbf{b}\|^2 + \mu \|\mathbf{x}\|^2 \right). \tag{18}$$

Either singular value decomposition, QR decomposition, or normal equations solve this optimization problem. We have chosen the last one due to its lower computational complexity.

## 6. Online Obstacle Avoidance and Re-Planning

After generating the trajectory, each UAV goes to its starting position and follows the planned path. During this process, they inform the others about cells that were explored and periodically check for obstacles within their path. Whenever the UAV detects an obstacle, it immediately stops, observes it for $t_o$ seconds, and then classifies the object as static if its position does not change during the waiting time or dynamic otherwise.

**Definition 8.** *An obstacle is the representation of any object that blocks the UAV from entering a cell. The robot is not allowed to explore a cell that is within an obstacle. Thus, a cell that contains an obstacle is considered by our algorithm as visited on the first run of the algorithm that the robot performs.*

**Definition 9.** *A moving obstacle is the representation of any object that temporarily blocks the UAV from entering a cell. A moving obstacle is only considered after the resulting path is generated. The moving obstacle is considered to be aware of its surroundings (e.g., an autonomous robot, a human, an animal, etc.).*

As above stated, as the moving obstacles are aware of their surroundings, the UAV does not need to consider an obstacle-avoiding procedure. Upon the moving obstacle detection, the UAV suspends the execution of the trajectory tracking temporarily and waits for the moving obstacle to pass and the pass to be clear.

Alternatively, the appearance of a new static obstacle triggers the information exchange between the robots and the suspension of the current trajectory execution. After including it on the map, each agent replans its trajectory with the new obstacle, revising the exploration status of each cell and re-executing the coverage algorithm.

## 7. Quantitative Tests

Multiple quantitative tests were performed to validate our approach. We performed mainly two quantitative tests: improvement of the environment partition, and evaluation of the trajectory smoothing. All quantitative tests were performed numerically in Matlab 2020a.

### 7.1. Improvement of the Environment Partition

First, we have launched the original (AWSTC [6]) and modified (IAWSTC) versions on 1000 worlds with dimensions $10 \times 10$ cells, on 500 worlds with dimensions $20 \times 20$ cells, and on 200 worlds with dimensions $30 \times 30$ cells with 2, 4, and 8 agents. The worlds and the robot starting points were generated in a random fashion.

**Definition 10.** *The redundancy ratio $\zeta_r$ is a metric that describes the coverage redundancy of the resulting trajectory. he redundancy ratio $\zeta_r$ is as follows:*

$$\zeta_r = \frac{\sum\limits_{i=1}^{N} \text{card}(S_i)}{\text{card}(E_{free})}, \tag{19}$$

*where* $\text{card}(S_i)$ *is the number of cells within the ith region* $S_i$, *N is the number of robots, and* $\text{card}(E_{free})$ *is the number of cells from the environment representation where there is no obstacle.*

**Definition 11.** *The ratio of equality* $\zeta_e$ *is a metric that measures the difference between the sizes of the subregions corresponding to each individual robot. This metric can be calculated as:*

$$\zeta_e = \frac{\max\limits_{i\in\{1,\dots,N\}} \text{card}(S_i)}{\frac{1}{N} \text{card}(E_{free})}, \tag{20}$$

**Definition 12.** *The increased length ratio* $\zeta_i$ *is a metric that defines the growth of the length of the resulting trajectory due to the smoothing step. It can be estimated as:*

$$\zeta_i = \frac{\text{length}(\tau_s)}{\text{length}(\tau)}, \tag{21}$$

*in which* $\text{length}(\tau_s)$ *is the smoothed trajectory length* $\tau_s$, *and* $\text{length}(\tau)$ *is the original trajectory length* $\tau$.

We used both the redundancy ratio $\zeta_r$ and the equality ratio $\zeta_e$ as metrics for the comparison. As shown in Figure 5, the IAWSTC algorithm improves both ratios.

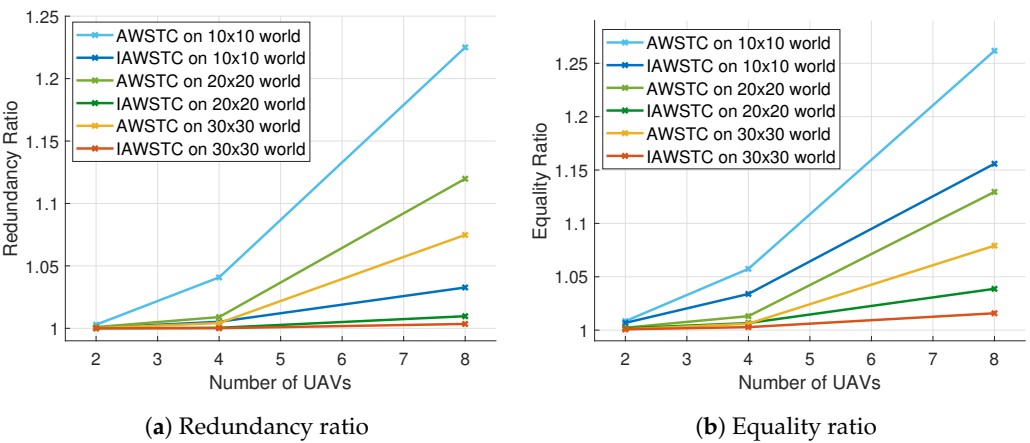

(**a**) Redundancy ratio        (**b**) Equality ratio

**Figure 5.** The relation between the redundancy ratio and equality ratio with respect to the number of drones and world size.

### 7.2. Evaluation of the Trajectory Smoothing

We also analyzed the performance of our TSULSS algorithm with respect to the values given to the $p$ and $\mu$ parameters. We tested an average of 100 random worlds with different sizes, such that $s \in \{10 \times 10, 11 \times 11, \dots, 25 \times 25\}$ with the number of exploring robots varying according to $N \in \{2, 3, \dots, 6\}$.

**Definition 13.** *The turn intensity* $\zeta_t$ *is a metric that specifies the intensity of the smoothness of each turn present in the resulting trajectory. An increase in turn intensity means an increase in the sharpness of the turns. It can be calculated as:*

$$\zeta_t = \frac{\sqrt{\sum\limits_{i=1}^{M-2}\left(\arccos\left(\frac{v_i \cdot v_{i+1}}{|v_i|\,|v_{i+1}|}\right)\right)^2}}{M-2}, \tag{22}$$

*in which M is the number of points* $P_1, \ldots, P_M \in \tau$. *Furthermore,* $v_i, v_{i+1}$ *can be calculated as:*

$$v_i = P_{i+1} - P_i \tag{23}$$

$$v_{i+1} = P_{i+2} - P_{i+1} \tag{24}$$

*Finally, this definition can be extended to consider closed trajectories. Thus, we append points* $P_{M+1} = P_1$ *and* $P_{M+2} = P_2$ *to the end of the resulting trajectory.*

We used the ratio of length increase $\zeta_i$, and the turn intensity $\zeta_t$, as indicated in Figure 6. We verified that the results proved that the higher the value of either the $\mu$ or the $p$ parameter, the smoother will be the trajectory and the shorter it will be its length. We also saw that the contrary is also valid.

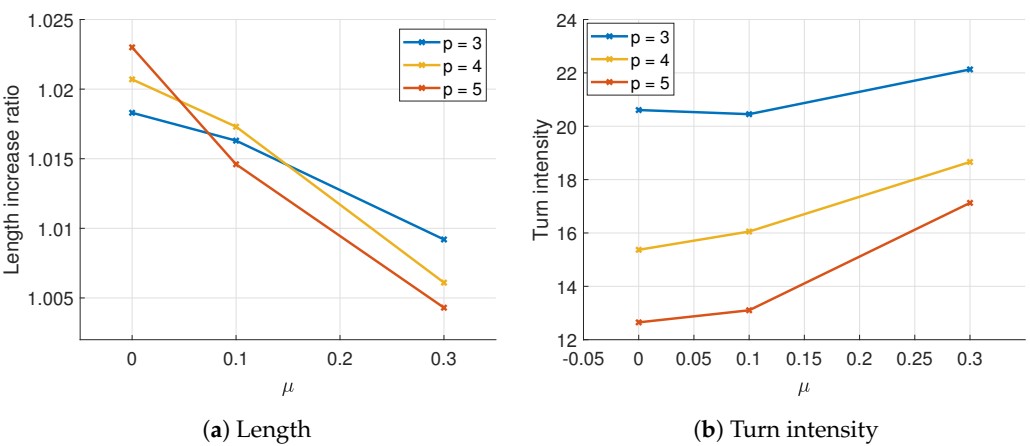

(**a**) Length  (**b**) Turn intensity

**Figure 6.** The effect of the increase on the length ratio and turn intensity on the parameters $p$ and $\mu$.

## 8. Results

We performed simulations using the realistic simulator of robotic operating system (ROS) and Gazebo [28]. The parameter values used in the simulation and real-world experiments are presented in Table 1.

**Table 1.** Used values of the parameters of our proposed algorithms.

|  | **Parameter** | **Value** |
|---|---|---|
| | $\sigma_a$ | 1 |
| | $\sigma_b$ | 100 |
| Partition of the Environment | $\sigma_d$ | 10 |
| | $\mathcal{E}_M$ | 10,000 |
| Smoothing | $p$ | 5 |
| | $\mu$ | 0.15 |
| Replanning | $t_o$ | 2 [s] |

### 8.1. Simulation

In this subsection we present the results from both numeric and realistic simulations. We performed several simulations on a computer with Intel Core i7, 8 GB RAM. Two case studies of numerical simulations are presented and three other cases of realistic gazebo simulations are also further presented.

#### 8.1.1. Numeric Simulation

The numeric simulations aimed to demonstrate the robustness of our approach with a larger environment and with more UAVs. These simulations used a world with a size of

93 m × 93 m, resulting in a 31 × 31 cell grid. From the several performed simulations, we present here two special cases. The first special case used 5 UAVs with a total of 15 fixed obstacles known a priori Figure 7a. The second special case used 7 UAVs with a total of 4 fixed obstacles known a priori Figure 7b. Note that in both cases there is an almost even distribution of regions allowing the coverage of the large area to be performed much easier.

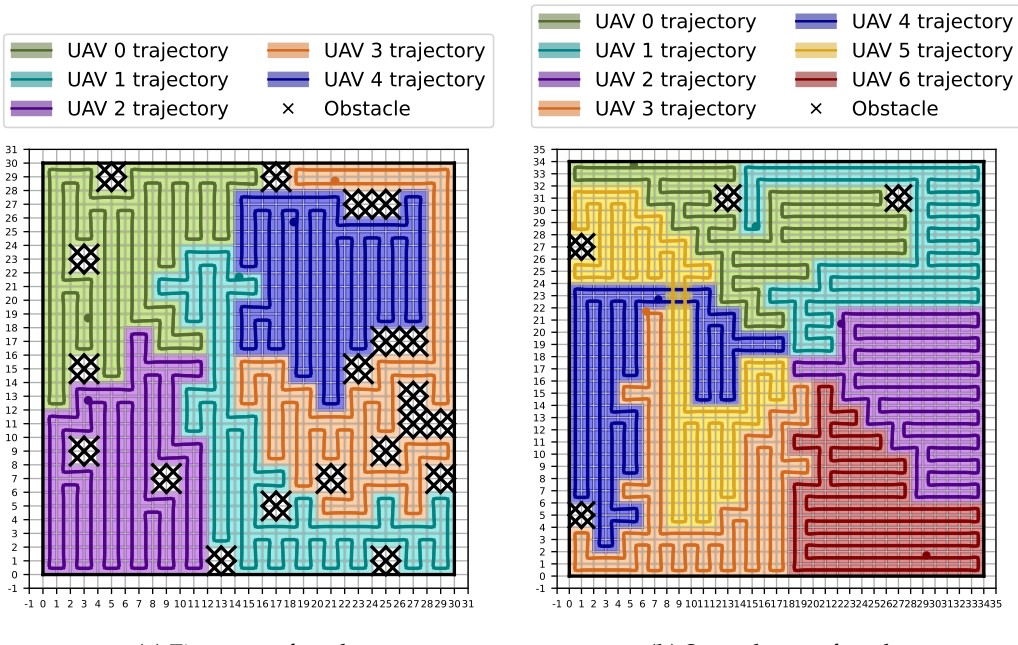

(**a**) First case of study       (**b**) Second case of study

**Figure 7.** Numerical simulations with large worlds.

8.1.2. Realistic Simulation

The three realistic simulations were performed using the MRS system [4] and the ROS/Gazebo realistic simulator. We used three drones in the first realistic simulation and two drones in the other two cases of study to validate our approach. In this first simulation, the world explored by the UAVs contained fixed obstacles known a priori. The simulated world had a size of 45 m × 45 m, resulting in a 15 × 15 cell grid. The trajectories resulting from the IAWSTC algorithm are shown in Figure 8.

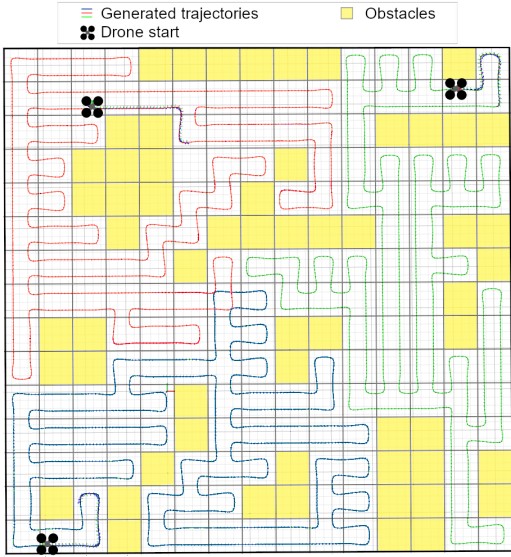

**Figure 8.** IAWSTC algorithm in simulation.

In this first simulation we had 13 fixed known obstacles that occupied the grid spaces seen in yellow in the figure above. These were a priori known obstacles. After the generated trajectory was performed by our approach, the complete coverage was achieved by the UAVs.

In the second realistic simulation, the two UAVs begin their fly in a world with a 30 m × 30 m of size, resulting in a 10 × 10 grid. This world had 8 fixed obstacles known a priori. The UAVs started moving according to the resulting trajectory (Figure 9a). During the simulation, one of the UAVs encounter a fixed obstacle that was unknown a priori. This obstacle was detected using sensors onboard the UAV. After detection, the UAV stops its movement and waits 5 s to check if it is moving or stationary. As UAV detected that it was a fixed obstacle, both UAVs stopped so that our approach could replan the whole coverage giving new trajectories to both UAVs (Figure 9b).

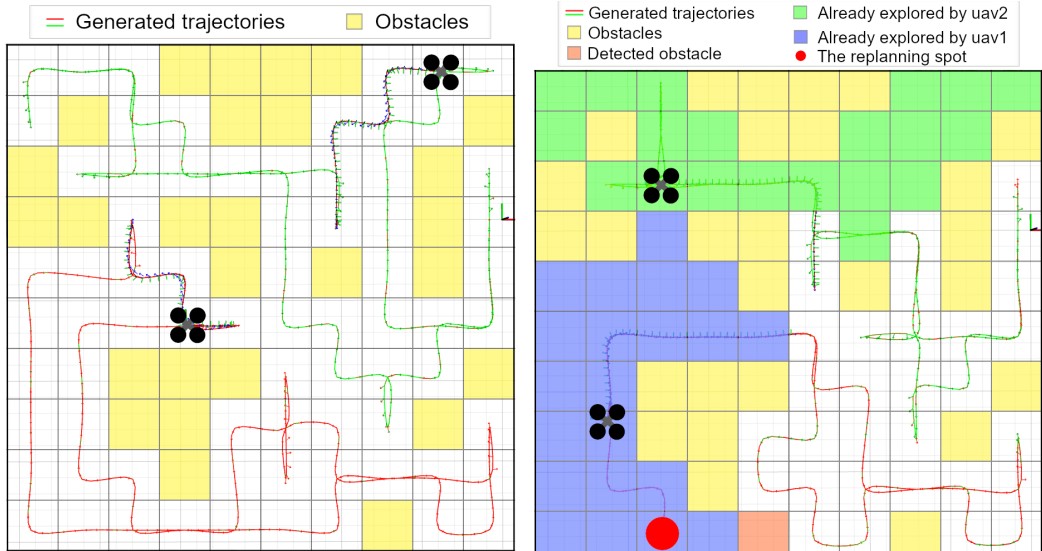

(**a**) Before encountering the fixed obstacle       (**b**) After encountering the fixed obstacle

**Figure 9.** Realistic Simulation with unknown fixed obstacle. VIDEO : https://youtu.be/4wocoYXDiz4, accessed on 17 December 2022.

In the third realistic simulation, the two UAVs begin their fly in a world with a 30 m × 30 m of size, resulting in a 10 × 10 grid. This world had 2 fixed obstacles known a priori. The UAVs started moving according to the resulting trajectory (Figure 10a). During the movement of the UAVs, a moving obstacle appears crossing the path of one of the UAVs (Figure 10b). Once more the obstacle is detected using sensors onboard the UAV. After detection, the UAV stops its movement and waits 5 s to check if it is moving or stationary. As it is a moving obstacle, eventually it moves out of the path of the UAV, and thus the UAV resumes its movement to track the planned trajectory (Figure 10c). The object is moving in an oscillating manner, going forward and back and the path of the obstacle is given by the blue line in the figures. Even with this oscillating movement, after the object clears the path the drone restarts to track the planned trajectory (Figure 10d).

### 8.2. Real Robot Experiments

We also performed real robot experiments with three UAVs based on the F450 frame. The UAVs have a GPS receiver for localization, 2D RPLiDAR for detecting obstacles, an Intel i7-based onboard computer, and a down-facing Garmin LIDAR for altitude estimation.

The IAWSTC was validated on a field with 40 m × 40 m of area. This resulted in a 10 × 10 cell grid. Some snapshots of the real robot experiments can be seen in Figure 11.

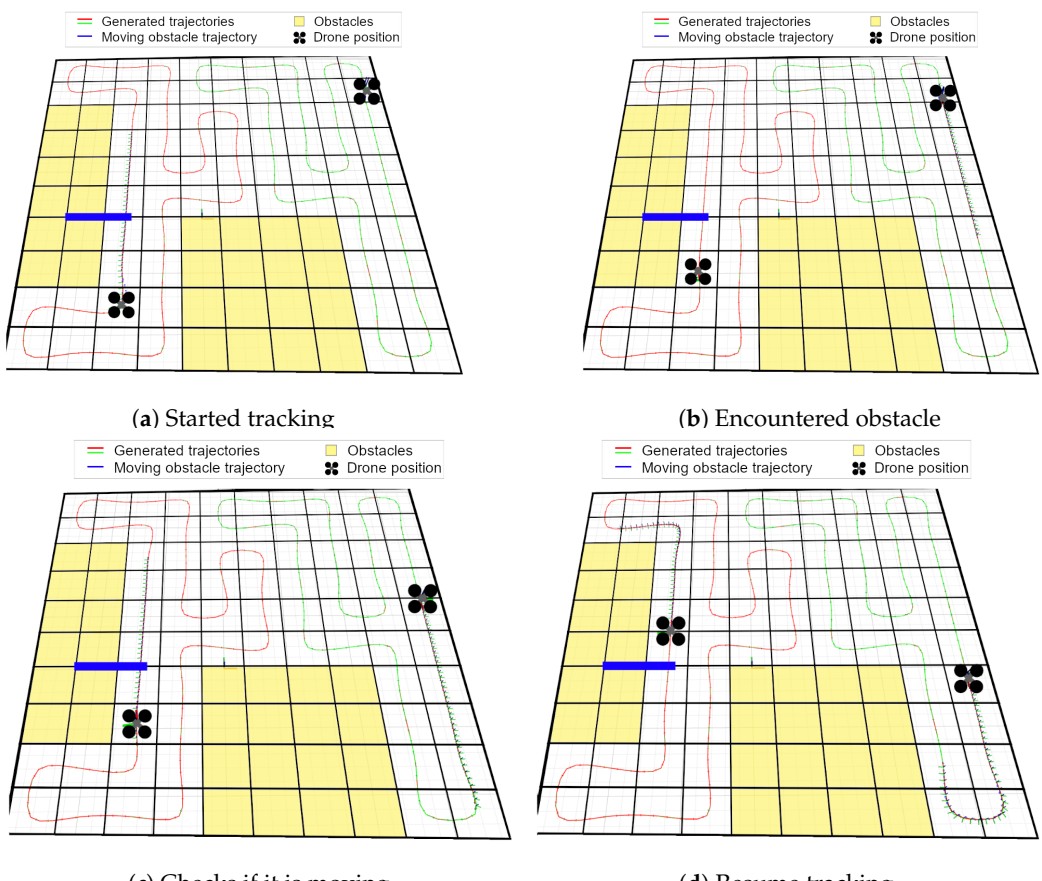

(**a**) Started tracking

(**b**) Encountered obstacle

(**c**) Checks if it is moving

(**d**) Resume tracking

**Figure 10.** Realistic Gazebo Simulation with unknown moving obstacle. VIDEO: https://youtu.be/KP-xCtVwBQs, accessed on 17 December 2022.

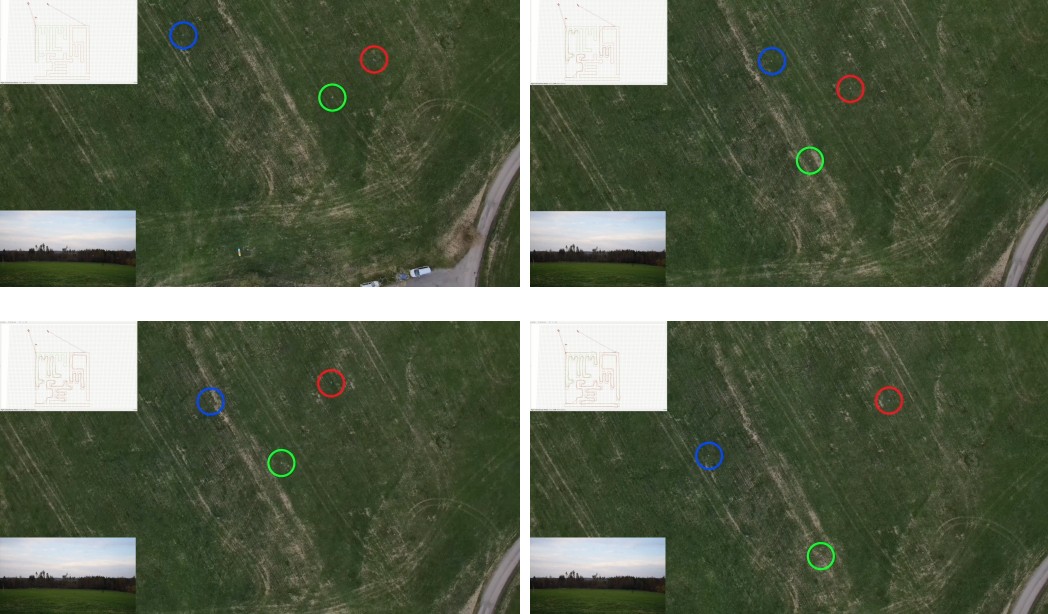

**Figure 11.** Snapshots of the performed real robot experiment. VIDEO: https://youtu.be/EjyLa0kJFaE, accessed on 17 December 2022.

We set the cell size to a 4 m × 4 m area so that the mechanism to predict collision with obstacles would not affect the resulting trajectories. The results from the implementation of IAWSTC are presented in Figure 12.

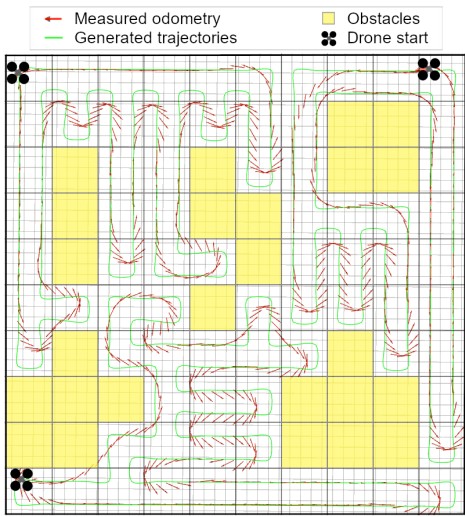

**Figure 12.** IAWSTC algorithm on real UAVs.

In the real robot experiments we had 5 fixed known obstacles that occupied the grid spaces seen in yellow in the figure above. These were also a priori known obstacles. After the generated trajectory was performed by our approach, the complete coverage was achieved by the UAVs.

## 9. Conclusions

In this work, we proposed an improvement of the AWSTC algorithm proposed by Dong et al. [6]. As a second contribution, we introduced an algorithm for smoothing with an online re-planner that avoids detected unexpected obstacles. Our modified version of the algorithm produces subregions with less redundancy than the original paper, using a mechanism to reduce intersections on multiple trajectories and with the ability to handle unexpected obstacles.

We performed experiments in simulation and also using real drones. The results of these tests verified that the IAWSTC algorithm produces subregions with less redundancy than its previous version. Our proposed approach can be applied in environment exploration in cluttered regions with the possibility of having dynamic obstacles. Further works are to perform a sequence of tests in real scenarios and statistically analyze the coverage rate using images acquired from the environments.

**Author Contributions:** Software, J.C.; experiments, J.C.; writing, T.A. and A.M.N.; writing–revision, supervision and project administration, T.P.N. All authors have read and agreed to the published version of the manuscript.

**Funding:** This work was supported by CTU grant no SGS20/174/OHK3/3T/13, by the Ministry of Education of the Czech Republic has funded my research by OP VVV funded project CZ.02.1.01/0.0/0.0/16 019/0000765 "Research Center for Informatics", and by the Technology Innovation Institute—Sole Proprietorship LLC, UAE, under the Research Project Contract No. TII/ARRC/2055/2021.

**Informed Consent Statement:** Not applicable.

**Data Availability Statement:** No new data were created or analyzed in this study. Data sharing is not applicable to this article.

**Conflicts of Interest:** The authors declare no conflict of interest.

## Abbreviations

The following abbreviations are used in this manuscript:

| | |
|---|---|
| AWSTC | artificially weighted spanning tree coverage |
| IAWSTC | improved artificially weighted spanning tree coverage |
| STC | spanning tree coverage |
| CPP | coverage path planning |
| MCPP | multi-robot coverage path planning |
| MAV | micro air vehicle |
| UAV | unmanned aerial vehicle |
| ROS | robotic operating system |
| TSULSS | trajectory smoothing using least squares solution |

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
