# Peer review of "An Improved Spanning Tree-Based Algorithm for Coverage of Large Areas Using Multi-UAV Systems"

_drones, doi:10.3390/drones7010009_

Round 1
Reviewer 1 Report
The authors claim to propose an improved spanning-tree method to address the problem of coverage path planning. The authors also present simulation and experiment results to verify the proposed method. However, this paper is not innovative enough to be accepted by the journal and may be more suitable for a conference article. The detailed comments are as follows.
- The innovation of current approach is not sufficiently justified. This work is a further improvement on Ref. [6]. The author should detail the innovation and contribution, and show exactly what improvements have been made.
- The trajectory smoothing needs further improving. In current experiment, the effect of UAVs tracking smooth route is very general. The authors should focus on UAV’s dynamics in trajectory smoothing.
- Section 1 to 3 are not clearly stated. In Section 1, authors highlight the difference between single UAV and multiple UAVs, but ignore the challenges of coverage path planning studied in the paper. In Section 2, authors state related works in the most straightforward manner, and do not have their own reflections and summaries. It is suggested that authors categorize the existing works to elaborate them in a sequential and progressive manner. In Section 3, authors should formulate the research problem and basic preliminaries used later rather than 16 trivial definitions. In fact, some definitions are better described by figures, such as Definitions 1 to 7, and some definitions can be presented where needed, such as Definitions 13 to 16.
- More simulation and experiment need to be added. The authors should give simulation results for multiple trails, as well as more specific experimental snapshots, and the corresponding analysis of the results.
Reviewer 2 Report
The authors proposed IAW-STC algorithm for environment exploration in cluttered regions, where unexpected obstacles were detected. However, more information is required in the results such as:
1. What is the number of obstacles? In case of dynamic obstacle, the motion of such obstacle must be observed in results (Fig. 4 and 5).
2. The authors performed outdoor experiments without obstacles. So, it is recommended to do experiments in the presence of real obstacles.
3. The author discussed clutters, but, how clutters were produced? It must also be shown in Figure 4 or 5.
4. How to identify number of drones in Figure 4? For clear presentation, there should be legend indicating each moving drone.
5. How much is the error in altitude and position? please plot an error diagram to show the accuracy of the proposed algorithm in comparison.
6. The authors stated that "We verified that the results proved that the higher the value of either the p or µ parameter, the smoother will be the trajectory and the bigger it will be its length". However, figure 3a shows opposite statement: at µ=0.3, p=5, the length increase ratio is below 1.005, it is higher at lower value of µ.
7. Smoothing means refining the estimates using more information about the desired trajectory (such as discussed in the references that I have recommended). However, the authors have not discussed more detail about smoothing and least squares. although the main heading is the smoothing least squares ...
8. the references are quite old and most of the recent works were cited from conferences. it is recommended to include some new recent works such as:
1. Detection and tracking of the trajectories of dynamic UAVs in restricted and cluttered environment (2021)
2. Dynamic based trajectory estimation and tracking in an uncertain environment (2021)
Round 2
Reviewer 1 Report
All comments have been revised and the current format can be accepted.
Reviewer 2 Report
I really appreciate the hard work and contribution of the authors. They have responded to all comments satisfactorily. Therefore, I recommend this paper be accepted in its current form.
Thanks